# THE LOSS SURFACE OF RESIDUAL NETWORKS: ENSEMBLES & THE ROLE OF BATCH NORMALIZATION

**Etai Littwin & Lior Wolf**
The School of Computer Science
Tel Aviv University, Israel
`{etailittwin,liorwolf}@gmail.com`

## ABSTRACT

Deep Residual Networks present a premium in performance in comparison to conventional networks of the same depth and are trainable at extreme depths. It has recently been shown that Residual Networks behave like ensembles of relatively shallow networks. We show that these ensembles are dynamic: while initially the virtual ensemble is mostly at depths lower than half the network's depth, as training progresses, it becomes deeper and deeper. The main mechanism that controls the dynamic ensemble behavior is the scaling introduced, e.g., by the Batch Normalization technique. We explain this behavior and demonstrate the driving force behind it. As a main tool in our analysis, we employ generalized spin glass models, which we also use in order to study the number of critical points in the optimization of Residual Networks.

## 1 INTRODUCTION

Residual Networks (He et al., 2015) (ResNets) are neural networks with skip connections. These networks, which are a specific case of Highway Networks (Srivastava et al., 2015), present state of the art results in the most competitive computer vision tasks including image classification and object detection.

The success of residual networks was attributed to the ability to train very deep networks when employing skip connections (He et al., 2016). A complementary view is presented by Veit et al. (2016), who attribute it to the power of ensembles and present an unraveled view of ResNets that depicts ResNets as an ensemble of networks that share weights, with a binomial depth distribution around half depth. They also present experimental evidence that short paths of lengths shorter than half-depth dominate the ResNet gradient during training.

The analysis presented here shows that ResNets are ensembles with a dynamic depth behavior. When starting the training process, the ensemble is dominated by shallow networks, with depths lower than half-depth. As training progresses, the effective depth of the ensemble increases. This increase in depth allows the ResNet to increase its effective capacity as the network becomes more and more accurate.

Our analysis reveals the mechanism for this dynamic behavior and explains the driving force behind it. This mechanism remarkably takes place within the parameters of Batch Normalization (Ioffe & Szegedy, 2015), which is mostly considered as a normalization and a fine-grained whitening mechanism that addresses the problem of internal covariate shift and allows for faster learning rates.

We show that the scaling introduced by batch normalization determines the depth distribution in the virtual ensemble of the ResNet. These scales dynamically grow as training progresses, shifting the effective ensemble distribution to bigger depths.

The main tool we employ in our analysis is spin glass models. Choromanska et al. (2015a) have created a link between conventional networks and such models, which leads to a comprehensive study of the critical points of neural networks based on the spin glass analysis of Auffinger et al. (2013). In our work, we generalize these results and link ResNets to generalized spin glass models. These models allow us to analyze the dynamic behavior presented above. Finally, we apply the results of Auffinger & Arous (2013) in order to study the loss surface of ResNets.

## 2 A RECAP OF CHOROMANSKA ET AL. (2015A)

We briefly summarize Choromanska et al. (2015a), which connects the loss function of multilayer networks with the hamiltonian of the p spherical spin glass model, and state their main contributions and results. The notations of our paper are summarized in Appendix A and slightly differ from those in Choromanska et al. (2015a).

A simple feed forward fully connected network $\mathcal{N}$, with $p$ layers and a single output unit is considered. Let $n_i$ be the number of units in layer $i$, such that $n_0$ is the dimension of the input, and $n_p = 1$. It is further assumed that the ReLU activation functions denoted by $\mathcal{R}()$ are used. The output $Y$ of the network given an input vector $x \in R^d$ can be expressed as

$$Y = \sum_{i=1}^{d} \sum_{j=1}^{\gamma} x_{ij} A_{ij} \prod_{k=1}^{p} w_{ij}^{(k)}, \tag{1}$$

where the first summation is over the network inputs $x_1...x_d$, and the second is over all paths from input to output. There are $\gamma = \prod_{i=1}^{p} n_i$ such paths and $\forall i, \quad x_{i1} = x_{i2} = ...x_{i\gamma}$. The variable $A_{ij} \in \{0,1\}$ denotes whether the path is active, i.e., whether all of the ReLU units along this path are producing positive activations, and the product $\prod_{k=1}^{p} w_{ij}^{(k)}$ represents the specific weight configuration $w_{ij}^1...w_{ij}^k$ multiplying $x_i$ given path $j$. It is assumed throughout the paper that the input variables are sampled i.i.d from a normal Gaussian distribution.

**Definition 1.** *The mass of the network $\mathcal{N}$ is defined as $\psi = \prod_{i=0}^{p} n_i$.*

The variables $A_{ij}$ are modeled as independent Bernoulli random variables with a success probability $\rho$, i.e., each path is equally likely to be active. Therefore,

$$\mathbb{E}_A[Y] = \sum_{i=1}^{d} \sum_{j=1}^{\gamma} x_{ij} \rho \prod_{k=1}^{p} w_{ij}^{(k)}. \tag{2}$$

The task of binary classification using the network $\mathcal{N}$ with parameters $\mathbf{w}$ is considered, using either the hinge loss $\mathcal{L}_{\mathcal{N}}^h$ or the absolute loss $\mathcal{L}_{\mathcal{N}}^a$:

$$\mathcal{L}_{\mathcal{N}}^h(\boldsymbol{w}) = \mathbb{E}_A[max(0, 1 - Y_x Y)], \quad \mathcal{L}_{\mathcal{N}}^a(\boldsymbol{w}) = \mathbb{E}_A[|Y_x - Y|] \tag{3}$$

where $Y_x$ is a random variable corresponding to the true label of sample $x$. In order to equate either loss with the hamiltonian of the p-spherical spin glass model, a few key approximations are made:

**A1** Variable independence - The inputs $x_{ij}$ are modeled as independent normal Gaussian random variables.

**A2** Redundancy in network parameterization - It is assumed the set of all the network weights $[w_1, w_2...w_N]$ contains only $\Lambda$ unique weights such that $\Lambda < N$.

**A3** Uniformity - It is assumed that all unique weights are close to being evenly distributed on the graph of connections defining the network $\mathcal{N}$. Practically, this means that we assume every node is adjacent to an edge with any one of the $\Lambda$ unique weights.

**A4** Spherical constraint - The following is assumed:

$$\frac{1}{\Lambda} \sum_{i=1}^{\Lambda} w_i^2 = C^2 \tag{4}$$

for some constant $C > 0$.

These assumptions are made for the sake of analysis, and do not necessarily hold. The validity of these assumption was posed as an open problem in Choromanska et al. (2015b), where a different degree of plausibility was assigned to each. Specifically, **A1**, as well as the independence assumption of $A_{ij}$, were deemed unrealistic, and **A2** - **A4** as plausible. For example, **A1** does not hold since each input $x_i$ is associated with many different paths and $x_{i1} = x_{i2} = ...x_{i\gamma}$. See Choromanska et al. (2015a) for further justification of these approximations.

Under **A1**–**A4**, the loss takes the form of a centered Gaussian process on the sphere $S^{\Lambda-1}(\sqrt{\Lambda})$. Specifically, it is shown to resemble the hamiltonian of the a spherical p-spin glass model given by:

$$\mathcal{H}_{p,\Lambda}(\tilde{\boldsymbol{w}}) = \frac{1}{\Lambda^{\frac{p-1}{2}}} \sum_{i_1...i_p}^{\Lambda} x_{i_1...i_p} \prod_{k=1}^{r} \tilde{w}_{i_k} \tag{5}$$

with spherical constraint

$$\frac{1}{\Lambda} \sum_{i=1}^{\Lambda} \tilde{w}_i^2 = 1 \tag{6}$$

where $x_{i_1...i_p}$ are independent normal Gaussian variables.

In Auffinger et al. (2013), the asymptotic complexity of spherical p spin glass model is analyzed based on random matrix theory. In Choromanska et al. (2015a) these results are used in order to shed light on the optimization process of neural networks. For example, the asymptotic complexity of spherical spin glasses reveals a layered structure of low-index critical points near the global optimum. These findings are then given as a possible explanation to several central phenomena found in neural networks optimization, such as similar performance of large nets, and the improbability of getting stuck in a "bad" local minima.

As part of our work, we follow a similar path. First, a link is formed between residual networks and the hamiltonian of a general multi-interaction spherical spin glass model as given by:

$$\mathcal{H}_{p,\Lambda}(\tilde{\boldsymbol{w}}) = \sum_{r=1}^{p} \frac{\epsilon_r}{\Lambda^{\frac{r-1}{2}}} \sum_{i_1,i_2...i_r=1}^{\Lambda} x_{i_1,i_2...i_r} \prod_{k=1}^{r} \tilde{w}_{i_k} \tag{7}$$

where $\epsilon_1...\epsilon_p$ are positive constants. Then, using Auffinger & Arous (2013), we obtain insights on residual networks. The other part of our work studies the dynamic behavior of residual networks, where we relax the assumptions made for the spin glass model.

## 3 RESIDUAL NETS AND GENERAL SPIN GLASS MODELS

We begin by establishing a connection between the loss function of deep residual networks and the hamiltonian of the general spherical spin glass model. We consider a simple feed forward fully connected network $\mathcal{N}$, with ReLU activation functions and residual connections. For simplicity of notations without the loss of generality, we assume $n_1 = ... = n_p = n$. $n_0 = d$ as before. In our ResNet model, there exist $p-1$ identity connections skipping a single layer each, starting from the first hidden layer. The output of layer $l > 1$ is given by:

$$\mathcal{N}_l(x) = \mathcal{R}(W_l^\top \mathcal{N}_{l-1}(x)) + \mathcal{N}_{l-1}(x) \tag{8}$$

where $W_l$ denotes the weight matrix connecting layer $l-1$ with layer $l$. Notice that the first hidden layer has no parallel skip connection, and so $\mathcal{N}_1(x) = \mathcal{R}(W_1^\top x)$. Without loss of generality, the scalar output of the network is the sum of the outputs of the output layer $p$ and is expressed as

$$Y = \sum_{r=1}^{p} \sum_{i=1}^{d} \sum_{j=1}^{\gamma_r} x_{ij}^{(r)} A_{ij}^{(r)} \prod_{k=1}^{r} w_{ij}^{(r)(k)} \tag{9}$$

where $A_{ij}^{(r)} \in \{0,1\}$ denotes whether path $j$ of length $r$ is open, and $\forall j, j', r, r'\ x_{ij}^r = x_{ij'}^{r'}$. The residual connections in $\mathcal{N}$ imply that the output $Y$ is now the sum of products of different lengths, indexed by $r$. Since our ResNet model attaches a skip connection to every layer except the first, $1 \leq r \leq p$. See Sec. 6 regarding models with less frequent skip connections.

Each path of length $r$ includes $r-1$ non-skip connections (those involving the first term in Eq. 8 and not the second, identity term) out of layers $l = 2..p$. Therefore, $\gamma_r = \binom{p-1}{r-1} n^r$. We define the following measure on the network:

**Definition 2.** *The mass of a depth $r$ subnetwork in $\mathcal{N}$ is defined as $\psi_r = d\gamma_r$.*

The properties of redundancy in network parameters and their uniform distribution, as described in Sec. 2, allow us to re-index Eq. 9.

**Lemma 1.** *Assuming assumptions* **A2** $-$ **A4** *hold, and* $\frac{n}{\Lambda} \in \mathbb{Z}$, *then the output can be expressed after reindexing as:*

$$Y = \sum_{r=1}^{p} \sum_{i_1,i_2...i_r=1}^{\Lambda} \sum_{j=1}^{\frac{\psi_r}{\Lambda^r}} x_{i_1,i_2...i_r}^{(j)} A_{i_1,i_2...i_r}^{(j)} \prod_{k=1}^{r} w_{i_k}. \tag{10}$$

*All proofs can be found in Appendix B.*

Making the modeling assumption that the ReLU gates are independent Bernoulli random variables with probability $\rho$, we obtain that for every path of length $r$, $\mathbb{E} A_{i_1,i_2...i_r}^{(j)} = \rho^r$ and

$$\mathbb{E}_A[Y] = \sum_{r=1}^{p} \sum_{i_1,i_2...i_r=1}^{\Lambda} \sum_{j=1}^{\frac{\psi_r}{\Lambda^r}} x_{i_1,i_2...i_r}^{(j)} \rho^r \prod_{k=1}^{r} w_{i_k}. \tag{11}$$

In order to connect ResNets to generalized spherical spin glass models, we denote the variables:

$$\xi_{i_1,i_2...i_r} = \sum_{j=1}^{\frac{\psi_r}{\Lambda^r}} x_{i_1,i_2...i_r}^{j}, \quad \tilde{x}_{i_1,i_2...i_r} = \frac{\xi_{i_1,i_2...i_r}}{\mathbb{E}_x[\xi_{i_1,i_2...i_r}^2]^{\frac{1}{2}}} \tag{12}$$

Note that since the input variables $x_1...x_d$ are sampled from a centered Gaussian distribution (dependent or not), then the set of variables $\tilde{x}_{i_1,i_2...i_r}$ are dependent normal Gaussian variables.

**Lemma 2.** *Assuming* **A2** $-$ **A3** *hold, and* $\frac{n}{\Lambda} \in \mathbb{N}$ *then* $\forall_{r,i_1...i_r}$ *the following holds:*

$$\frac{1}{d}(\frac{\psi_r}{\Lambda^r})^2 \le \mathbb{E}[\xi_{i_1,i_2...i_r}^2] \le (\frac{\psi_r}{\Lambda^r})^2. \tag{13}$$

We approximate the expected output $\boldsymbol{E}_A(Y)$ with $\hat{Y}$ by assuming the minimal value in 13 holds such that $\forall_{r,i_1...i_r} \quad \mathbb{E}[\xi_{i_1,i_2...i_r}^2] = \frac{1}{d}(\frac{\psi_r}{\Lambda^r})^2$. This approximation holds exactly when $\Lambda = n$, since all weight configurations of a particular length in Eq. 10 will appear the same number of times. When $\Lambda \neq n$, the uniformity assumption dictates that each configuration of weights would appear approximately equally regardless of the inputs, and the expectation values would be very close to the lower bound. The following expression for $\hat{Y}$ is thus obtained:

$$\hat{Y} = \sum_{r=1}^{p}(\frac{\rho}{\Lambda})^r \frac{\psi_r}{\sqrt{d}} \sum_{i_1,i_2...i_r=1}^{\Lambda} \tilde{x}_{i_1,i_2...i_r} \prod_{k=1}^{r} w_{i_k}. \tag{14}$$

The independence assumption **A1** was not assumed yet, and 14 holds regardless. Assuming **A4** and denoting the scaled weights $\tilde{w}_i = \frac{1}{C} w_i$, we can link the distribution of $\hat{Y}$ to the distribution on $\tilde{x}$:

$$\hat{Y} = \sum_{r=1}^{p} \frac{\psi_r}{\sqrt{d}}(\frac{\rho C}{\Lambda})^r \sum_{i_1,i_2...i_r=1}^{\Lambda} \tilde{x}_{i_1,i_2...i_r} \prod_{k=1}^{r} \tilde{w}_{i_k}$$

$$= z \sum_{r=2}^{p} \frac{\epsilon_r}{\Lambda^{\frac{r-1}{2}}} \sum_{i_1,i_2...i_r=1}^{\Lambda} \tilde{x}_{i_1,i_2...i_r} \prod_{k=1}^{r} \tilde{w}_{i_k} \tag{15}$$

where $\epsilon_r = \epsilon_r = \frac{1}{z}\binom{p-1}{r-1}(\frac{\rho n C}{\sqrt{\Lambda}})^r$ and $z$ is a normalization factor such that $\sum_{r=1}^{p} \epsilon_r^2 = 1$.

The following lemma gives a generalized expression for the binary and hinge losses of the network.

**Lemma 3** ( Choromanska et al. (2015a)). *Assuming assumptions* **A2** $-$ **A4** *hold, then both the losses* $\mathcal{L}_{\mathcal{N}}^{h}(x)$ *and* $\mathcal{L}_{\mathcal{N}}^{a}(x)$ *can be generalized to a distribution of the form:*

$$\mathcal{L}_{\mathcal{N}}(x) = C_1 + C_2 \hat{Y} \tag{16}$$

*where* $C_1, C_2$ *are positive constants that do not affect the optimization process.*

The model in Eq. 16 has the form of a spin glass model, except for the dependency between the variables $\tilde{x}_{i_1, i_2 \ldots i_r}$. We later use an assumption similar to **A1** of independence between these variables in order to link the two binary classification losses and the general spherical spin glass model. However, for the results in this section, this is not necessary.

We denote the important quantities:

$$\beta = \frac{\rho n C}{\sqrt{\Lambda}}, \quad \epsilon_r = \frac{1}{z} \binom{p-1}{r-1} \beta^r \tag{17}$$

The series $(\epsilon_r)_{r=1}^p$ determines the weight of interactions of a specific length in the loss surface. Notice that for constant depth $p$ and large enough $\beta$, $\arg\max_r(\epsilon_r) = p$. Therefore, for wide networks, where $n$ and, therefore, $\beta$ are large, interactions of order $p$ dominate the loss surface, and the effect of the residual connections diminishes. Conversely, for constant $\beta$ and a large enough $p$ (deep networks), we have that $\arg\max_r(\epsilon_r) < p$, and can expect interactions of order $r < p$ to dominate the loss. The asymptotic behavior of $\epsilon$ is captured by the following lemma:

**Theorem 1.** *Assuming $\frac{\beta}{1+\beta}p \in \mathbb{N}$, we have that:*

$$\lim_{p \to \infty} \frac{1}{p} \arg\max_r(\epsilon_r) = \frac{\beta}{1+\beta} \tag{18}$$

As the next theorem shows, the epsilons are concentrated in a narrow band near the maximal value.

**Theorem 2.** *For any $\alpha_1 < \frac{\beta}{1+\beta} < \alpha_2$, and assuming $\alpha_1 p, \alpha_2 p, \frac{\beta}{1+\beta}p \in \mathbb{N}$, it holds that:*

$$\lim_{p \to \infty} \sum_{r=\alpha_1 p}^{\alpha_2 p} \epsilon_r^2 = 1 \tag{19}$$

Thm. 2 implies that for deep residual networks, the contribution of weight products of order far away from the maximum $\frac{\beta}{1+\beta}p$ is negligible. The loss is, therefor, similar in complexity to that of an ensemble of potentially shallow conventional nets. The next Lemma shows that we can shift the effective depth to any value by simply controlling $C$.

**Lemma 4.** *For any integer $1 \leq k \leq p$ there exists a global scaling parameter $C$ such that $\arg\max_r(\epsilon_r(C)) = k$.*

A simple global scaling of the weights is, therefore, enough to change the loss surface, from an ensemble of shallow conventional nets, to an ensemble of deep nets. This is illustrated in Fig. 1(a-c) for various values of $\beta$. In a common weight initialization scheme for neural networks, $C = \frac{1}{\sqrt{n}}$ (Orr & Müller, 2003; Glorot & Bengio, 2010). With this initialization and $\Lambda = n$, $\beta = \rho$ and the maximal weight is obtained at less than half the network's depth $\lim_{p \to \infty} \arg\max_r(\epsilon_r) < \frac{p}{2}$. Therefore, at the initialization, the loss function is primarily influenced by interactions of considerably lower order than the depth $p$, which facilitates easier optimization.

## 4 DYNAMIC BEHAVIOR OF RESIDUAL NETS

The expression for the output of a residual net in Eq. 15 provides valuable insights into the machinery at work when optimizing such models. Thm. 1 and 2 imply that the loss surface resembles that of an ensemble of shallow nets (although not a real ensemble due to obvious dependencies), with various depths concentrated in a narrow band. As noticed in Veit et al. (2016), viewing ResNets as ensembles of relatively shallow networks helps in explaining some of the apparent advantages of these models, particularly the apparent ease of optimization of extremely deep models, since deep paths barely affect the overall loss of the network. However, this alone does not explain the increase in accuracy of deep residual nets over actual ensembles of standard networks. In order to explain the improved performance of ResNets, we make the following claims:

1. The distribution of the depths of the networks within the ensemble is controlled by the scaling parameter $C$.

2. During training, $C$ changes and causes a shift of focus from a shallow ensemble to deeper and deeper ensembles, which leads to an additional capacity.

3. In networks that employ batch normalization, $C$ is directly embodied as the scale parameter $\lambda$. The starting condition of $\lambda = 1$ offers a good starting condition that involves extremely shallow nets.

For the remainder of Sec.4, we relax all assumptions, and assume that at some point in time $\frac{1}{\Lambda} \sum_{i=1}^{\Lambda} w_i^2 = C^2$, and $\Lambda = N$. Using Eq. 9 for the output of the network $\hat{Y}$ in Lemma. 3, the loss can be expressed:

$$\mathcal{L}_\mathcal{N}(x, \boldsymbol{w}) = C_1 + C_2 \sum_{r=1}^{p} \sum_{i=1}^{d} \sum_{j=1}^{\gamma_r} x_{ij}^{(r)} A_{ij}^{(r)} \prod_{k=1}^{r} w_{ij}^{(r)(k)} \tag{20}$$

where $C_1, C_2$ are some constants that do not affect the optimization process. In order to gain additional insight into this dynamic mechanism, we investigate the derivative of the loss with respect to the scale parameter $C$. Using Eq. 9 for the output, we obtain:

$$\frac{\partial \mathcal{L}_\mathcal{N}(x, \boldsymbol{w})}{\partial C} = \frac{C_2}{C} \sum_{r=1}^{p} r \sum_{i=1}^{d} \sum_{j=1}^{\gamma_r} x_{ij}^{(r)} A_{ij}^{(r)} \prod_{k=1}^{r} w_{ij}^{(r)(k)} \tag{21}$$

Notice that the addition of a multiplier $r$ indicates that the derivative is increasingly influenced by deeper networks.

## 4.1 BATCH NORMALIZATION

Batch normalization has shown to be a crucial factor in the successful training of deep residual networks. As we will show, batch normalization layers offer an easy starting condition for the network, such that the gradients from early in the training process will originate from extremely shallow paths.

We consider a simple batch normalization procedure, which ignores the additive terms, has the output of each ReLU unit in layer $l$ normalized by a factor $\sigma_l$ and then is multiplied by some parameter $\lambda_l$. The output of layer $l > 1$ is therefore:

$$\mathcal{N}_l(x) = \frac{\lambda_l}{\sigma_l} \mathcal{R}(W_l^\top \mathcal{N}_{l-1}(x)) + \mathcal{N}_{l-1}(x) \tag{22}$$

where $\sigma_l$ is the mean of the estimated standard deviations of various elements in the vector $\mathcal{R}(W_l^\top \mathcal{N}_{l-1}(x))$. Furthermore, a typical initialization of batch normalization parameters is to set $\forall_l, \lambda_l = 1$. In this case, providing that units in the same layer have equal variance $\sigma_l$, the recursive relation $\mathbb{E}[\mathcal{N}_{l+1}(x)_j^2] = 1 + \mathbb{E}[\mathcal{N}_l(x)_j^2]$ holds for any unit $j$ in layer $l$. This, in turn, implies that the output of the ReLU units should have increasing variance $\sigma_l^2$ as a function of depth. Multiplying the weight parameters in deep layers with an increasingly small scaling factor $\frac{1}{\sigma_l}$, effectively reduces the influence of deeper paths, so that extremely short paths will dominate the early stages of optimization. We next analyze how the weight scaling, as introduced by batch normalization, provides a driving force for the effective ensemble to become deeper as training progresses.

## 4.2 THE DRIVING FORCE BEHIND THE SCALE INCREASE

The analysis below focuses on a single residual connection skipping a block of one or more layers. Since it holds for each block individually, it holds also for a residual network of multiple skipped blocks of arbitrary depth.

We consider a simple network of depth $p$, with a single residual connection skipping $p - m$ layers. We further assume that batch normalization is applied at the output of each ReLU unit as described in Eq. 22. We denote by $l_1...l_m$ the indices of layers that are not skipped by the residual connection, and $\hat{\lambda}_m = \prod_{i=1}^{m} \frac{\lambda_{l_i}}{\sigma_{l_i}}$, $\hat{\lambda}_p = \prod_{i=1}^{p} \frac{\lambda_i}{\sigma_i}$. Since every path of length $m$ is multiplied by $\hat{\lambda}_m$, and every path of length $p$ is multiplied by $\hat{\lambda}_p$, the expression for the loss can be expressed using Eq. 20 and

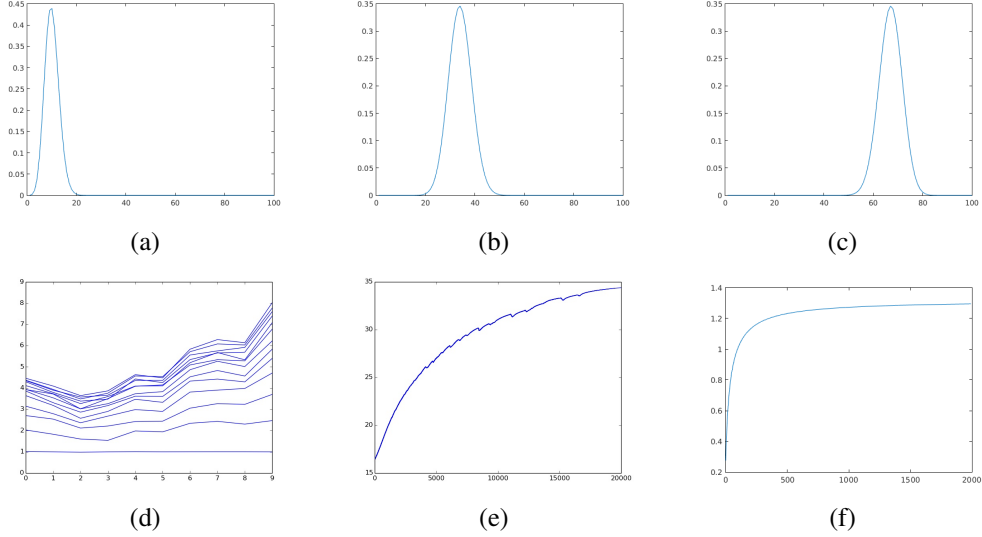

Figure 1: (a) A histogram of $\epsilon_r(\beta)$, $r = 1..p$, for $\beta = 0.1$ and $p = 100$ . (b) Same for $\beta = 0.5$ (c) Same for $\beta = 2$. (d) Values (y-axis) of the batch normalization parameters $\lambda_l$ (x-axis) for 10 layers ResNet trained to discriminate between 50 multivariate Gaussians (see Appendix C for more details). Higher plot lines indicate later stages of training. (e) The norm of the weights of a residual network, which does not employ batch normalization, as a function of the iteration. (f) The asymptotic of the mean number of critical points of a finite index as a function of $\beta$.

ignoring constant terms:

$$\mathcal{L}_{\mathcal{N}}(x, \boldsymbol{w}) = \hat{\lambda}_m \sum_{i=1}^{d} \sum_{j=1}^{\gamma_m} x_{ij}^{(m)} A_{ij}^{(m)} \prod_{k=1}^{r} w_{ij}^{(m)(k)} + \hat{\lambda}_p \sum_{i=1}^{d} \sum_{j=1}^{\gamma_p} x_{ij}^{(m)} A_{ij}^{(p)} \prod_{k=1}^{p} w_{ij}^{(p)(k)}$$

$$= \mathcal{L}_m(x, \boldsymbol{w}) + \mathcal{L}_p(x, \boldsymbol{w}) \quad (23)$$

We denote by $\nabla_{\boldsymbol{w}}$ the derivative operator with respect to the parameters $\boldsymbol{w}$, and the gradient $\boldsymbol{g} = \nabla_{\boldsymbol{w}} \mathcal{L}_{\mathcal{N}}(x, \boldsymbol{w}) = \boldsymbol{g}_m + \boldsymbol{g}_p$ evaluated at point $\boldsymbol{w}$.

**Theorem 3.** *Considering the loss in 23, and assuming $\frac{\partial \mathcal{L}_{\mathcal{N}}(x, \boldsymbol{w})}{\partial \lambda_l} = 0$, then for a small learning rate $0 < \mu << 1$ the following hold:*

1. *For any $\lambda_{l \in l_1 ... l_m}$ then:*

$$\left| \lambda_l - \mu \frac{\partial \mathcal{L}_{\mathcal{N}}(x, \boldsymbol{w} - \mu \boldsymbol{g})}{\partial \lambda_l} \right| > |\lambda_l| \quad (24)$$

2. *For any $\lambda_{l \notin l_1 ... l_m}$, if $\|\boldsymbol{g}_p\|_2^2 + \boldsymbol{g}_p^\top \boldsymbol{g}_m > 0$ then:*

$$\left| \lambda_l - \mu \frac{\partial \mathcal{L}_{\mathcal{N}}(x, \boldsymbol{w} - \mu \boldsymbol{g})}{\partial \lambda_l} \right| > |\lambda_l| \quad (25)$$

Thm. 3 suggests that $|\lambda_l|$ will increase for layers $l$ that do not have skip-connections. Conversely, if layer $l$ has a parallel skip connection, then $|\lambda_l|$ will increase if $\|\boldsymbol{g}_p\|_2 > \|\boldsymbol{g}_m\|_2$, where the later condition implies that shallow paths are nearing a local minima. Notice that an increase in $|\lambda_{l \notin l_1 ... l_m}|$ results in an increase in $|\tilde{\lambda}_p|$, while $|\tilde{\lambda}_m|$ remains unchanged, therefore shifting the balance into deeper ensembles.

This steady increase of $|\lambda_l|$, as predicted in our theoretical analysis, is also backed in experimental results, as depicted in Fig. 1(d). Note that the first layer, which cannot be skipped, behaves differently than the other layers. More experiments can be found in Appendix C.

It is worth noting that the mechanism for this dynamic property of residual networks can also be observed without the use of batch normalization, as a steady increase in the $L2$ norm of the weights, as shown in Fig. 1(e). In order to model this, consider the residual network as discussed above, without batch normalization layers. Recalling, $\|\boldsymbol{w}\|_2 = C\sqrt{\Lambda}, \tilde{\boldsymbol{w}} = \frac{\boldsymbol{w}}{C}$, the loss of this network is expressed as:

$$\mathcal{L}_{\mathcal{N}}(x, \boldsymbol{w}) = C^m \sum_{i=1}^{d} \sum_{j=1}^{\gamma_m} x_{ij}^{(m)} A_{ij}^{(m)} \prod_{k=1}^{r} \tilde{w}_{ij}^{(m)(k)} + C^p \sum_{i=1}^{d} \sum_{j=1}^{\gamma_p} x_{ij}^{(m)} A_{ij}^{(p)} \prod_{k=1}^{p} \tilde{w}_{ij}^{(p)(k)}$$
$$= \mathcal{L}_m(x, \boldsymbol{w}) + \mathcal{L}_p(x, \boldsymbol{w}) \quad (26)$$

**Theorem 4.** *Considering the loss in 26, and assuming $\frac{\partial \mathcal{L}_{\mathcal{N}}(x,\boldsymbol{w})}{\partial C} = 0$, then for a small learning rate $0 < \mu << 1$ the following hold:*

$$\frac{\partial \mathcal{L}_{\mathcal{N}}(x, \boldsymbol{w} - \mu\boldsymbol{g})}{\partial C} \approx -\mu \frac{1}{C}(m\|\boldsymbol{g}_m\|_2^2 + p\|\boldsymbol{g}_p\|_2^2 + (m+p)\boldsymbol{g}_p^\top \boldsymbol{g}_m) \quad (27)$$

Thm. 4 indicates that if either $\|\boldsymbol{g}_p\|_2$ or $\|\boldsymbol{g}_m\|_2$ is dominant (for example, near local minimas of the shallow network, or at the start of training), the scaling of the weights $C$ will increase. This expansion will, in turn, emphasize the contribution of deeper paths over shallow paths, and increase the overall capacity of the residual network. This dynamic behavior of the effective depth of residual networks is of key importance in understanding the effectiveness of these models. While optimization starts off rather easily with gradients largely originating from shallow paths, the overall advantage of depth is still maintained by the dynamic increase of the effective depth.

## 5   THE LOSS SURFACE OF ENSEMBLES

We now present the results of Auffinger & Arous (2013) regarding the asymptotic complexity in the case of $\lim_{\Lambda \to \infty}$ of the multi-spherical spin glass model given by:

$$\mathcal{H}_{\boldsymbol{\epsilon}, \Lambda} = -\sum_{r=2}^{\infty} \frac{\epsilon_r}{\Lambda^{\frac{r-1}{2}}} \sum_{i_1, \dots i_r = 1}^{\Lambda} J_{i_1 \dots i_r}^{r} \tilde{w}_{i_2} \dots \tilde{w}_{i_r} \quad (28)$$

where $J_{i_1 \dots i_r}^{r}$ are independent centered standard Gaussian variables, and $\boldsymbol{\epsilon} = (\epsilon_r)_{r>2}$ are positive real numbers such that $\sum_{r=2}^{\infty} \epsilon_r 2^r < \infty$. A configuration $\boldsymbol{w}$ of the spin spherical spin-glass model is a vector in $R^\Lambda$ satisfying the spherical constraint:

$$\frac{1}{\Lambda} \sum_{i=1}^{\Lambda} w_i^2 = 1, \quad \sum_{r=2}^{\infty} \epsilon_r^2 = 1 \quad (29)$$

Note that the variance of the process is independent of $\boldsymbol{\epsilon}$:

$$E[\mathcal{H}_{\boldsymbol{\epsilon}, \Lambda}^2] = \sum_{r=2}^{\infty} \Lambda^{1-r} \epsilon_r^2 (\sum_{i=1}^{\Lambda} w_i^2)^r = \Lambda \sum_{r=1}^{\infty} \epsilon_r^2 = \Lambda \quad (30)$$

**Definition 3.** *We define the following:*

$$v' = \sum_{r=2}^{\infty} \epsilon_r^2 r, \quad v'' = \sum_{r=2}^{\infty} \epsilon_r^2 r(r-1), \quad \alpha^2 = v'' + v' - v'^2 \quad (31)$$

Note that for the single interaction spherical spin model $\alpha^2 = 0$. The index of a critical point of $H_{\boldsymbol{\epsilon}, \Lambda}$ is defined as the number of negative eigenvalues in the hessian $\nabla^2 H_{\boldsymbol{\epsilon}, \Lambda}$ evaluated at the critical point $\boldsymbol{w}$.

**Definition 4.** *For any $0 \le k < \Lambda$ and $u \in \mathcal{R}$, we denote the random number $Crt_{\lambda,k}(u, \boldsymbol{\epsilon})$ as the number of critical points of the hamiltonian in the set $BX = \{\Lambda X | X \in (-\infty, u)\}$ with index $k$. That is:*

$$Crt_{\Lambda,k}(u, \boldsymbol{\epsilon}) = \sum_{\boldsymbol{w}: \nabla H_{\boldsymbol{\epsilon}, \Lambda} = 0} \mathbb{1}\{H_{\boldsymbol{\epsilon}, \Lambda} \in \Lambda u\} \mathbb{1}\{i(\nabla^2 H_{\boldsymbol{\epsilon}, \Lambda}) = k\} \quad (32)$$

Furthermore, define $\theta_k(u, \epsilon) = \lim_{\Lambda \to \infty} \frac{1}{\Lambda} log \, \mathbb{E}[Crt_{\Lambda,k}(u\epsilon)]$. Corollary 1.1 of Auffinger & Arous (2013) states that for any $k > 0$:

$$\theta_k(\mathbb{R}, \epsilon) = \frac{1}{2}log(\frac{v''}{v'}) - \frac{v'' - v'}{v'' + v'} \tag{33}$$

Eq. 33 provides the asymptotic mean total number of critical points with non-diverging index $k$. It is presumed that the SGD algorithm will easily avoid critical points with a high index that have many descent directions, and maneuver towards low index critical points. We, therefore, investigate how the mean total number of low index critical points vary as the ensemble distribution embodied in $(\epsilon_r)_{r>2}$ changes its shape by a steady increase in $\beta$.

Fig. 1(f) shows that as the ensemble progresses towards deeper networks, the mean amount of low index critical points increases, which might cause the SGD optimizer to get stuck in local minima. This is, however, resolved by the the fact that by the time the ensemble becomes deep enough, the loss function has already reached a point of low energy as shallower ensembles were more dominant earlier in the training. In the following theorem, we assume a finite ensemble such that $\sum_{r=p+1}^{\infty} \epsilon_r 2^r \approx 0$.

**Theorem 5.** *For any $k \in \mathbb{N}, p > 1$, we denote the solution to the following constrained optimization problems:*

$$\epsilon^* = \arg \max_{\epsilon} \theta_k(\mathbb{R}, \epsilon) \quad s.t \quad \sum_{r=2}^{p} \epsilon_r^2 = 1 \tag{34}$$

*It holds that:*

$$\epsilon_r^* = \begin{cases} 1, & r = p \\ 0, & otherwise \end{cases} \tag{35}$$

Thm. 5 implies that any heterogeneous mixture of spin glasses contains fewer critical points of a finite index, than a mixture in which only $p$ interactions are considered. Therefore, for any distribution of $\epsilon$ that is attainable during the training of a ResNet of depth $p$, the number of critical points is lower than the number of critical points for a conventional network of depth $p$.

# 6 DISCUSSION

In this work, we use spin glass analysis in order to understand the dynamic behavior ResNets display during training and to study their loss surface. In particular, we use at one point or another the assumptions of redundancy in network parameters, near uniform distribution of network weights, independence between the inputs and the paths and independence between the different copies of the input as described in Choromanska et al. (2015a). The last two assumptions, i.e., the two independence assumptions, are deemed in Choromanska et al. (2015b) as unrealistic, while the remaining are considered plausible.

Our analysis of critical points in ensembles (Sec. 5) requires all of the above assumptions. However, Thm. 1 and 2, as well as Lemma. 4, do not assume the last assumption, i.e., the independence between the different copies of the input. Moreover, the analysis of the dynamic behavior of residual nets (Sec. 4) does not assume any of the above assumptions.

Our results are well aligned with some of the results shown in Larsson et al. (2016), where it is noted empirically that the deepest column trains last. This is reminiscent of our claim that the deeper networks of the ensemble become more prominent as training progresses. The authors of Larsson et al. (2016) hypothesize that this is a result of the shallower columns being stabilized at a certain point of the training process. In our work, we discover the exact driving force that comes into play.

In addition, our work offers an insight into the mechanics of the recently proposed densely connected networks (Huang et al., 2016). Following the analysis we provide in Sec. 3, the additional shortcut paths decrease the initial capacity of the network by offering many more short paths from input to output, thereby contributing to the ease of optimization when training starts. The driving force mechanism described in Sec. 4.2 will then cause the effective capacity of the network to increase.

Note that the analysis presented in Sec. 3 can be generalized to architectures with arbitrary skip connections, including dense nets. This is done directly by including all of the induced sub networks in Eq. 9. The reformulation of Eq. 10 would still holds, given that $\Psi_r$ is modified accordingly.

## 7 Conclusion

Ensembles are a powerful model for ResNets, which unravels some of the key questions that have surrounded ResNets since their introduction. Here, we show that ResNets display a dynamic ensemble behavior, which explains the ease of training such networks even at very large depths, while still maintaining the advantage of depth. As far as we know, the dynamic behavior of the effective capacity is unlike anything documented in the deep learning literature. Surprisingly, the dynamic mechanism typically takes place within the outer multiplicative factor of the batch normalization module.

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

## A Summary of notations

Table 1 presents the various symbols used throughout this work and their meaning.

Table 1: Notations

| SYMBOL | DESCRIPTION |
|---|---|
| x | Input vector $\in \mathbb{R}^d$, sampled from a normal distribution |
| d | The dimensionality of the input $x$ |
| $\mathcal{N}_i(x)$ | The output of layer $i$ of network $\mathcal{N}$ given input $x$ |
| $Y$ | The final output of the network $\mathcal{N}$ |
| $Y_x$ | True label of input $x$ |
| $\mathcal{L}_{\mathcal{N}}$ | Loss function of network $\mathcal{N}$ |
| $\mathcal{L}_{\mathcal{N}}^h$ | Hinge loss |
| $\mathcal{L}_{\mathcal{N}}^a$ | Absolute loss |
| $p$ | The depth of network $\mathcal{N}$ |
| $\boldsymbol{w}$ | Weights of the network $\boldsymbol{w} \in \mathbb{R}^{\Lambda}$ |
| C | A positive scale factor such that $\|\boldsymbol{w}\|_2 = \sqrt{\Lambda}C$ |
| $\tilde{\boldsymbol{w}}$ | Scaled weights such that $\tilde{\boldsymbol{w}} = \frac{1}{C}\boldsymbol{w}$ |
| n | The number of units in layers $l > 0$ |
| $\Lambda$ | The number of unique weights in the network |
| $N$ | The total number of weights in the network $\mathcal{N}$ |
| $W_l$ | The weight matrix connecting layer $l-1$ to layer $l$ in $\mathcal{N}$. |
| $\mathcal{H}_{p,\lambda}$ | The hamiltonian of the $p$ interaction spherical spin glass model. |
| $\mathcal{H}_{\boldsymbol{\epsilon},\Lambda}$ | The hamiltonian of the general spherical spin glass model. |
| $\gamma$ | Total number of paths from input to output in network $\mathcal{N}$ |
| $\psi$ | $\gamma d$ |
| $\gamma_r$ | Total number of paths from input to output in network $\mathcal{N}$ of length $r$ |
| $\psi_r$ | $\gamma_r d$ |
| $\mathcal{R}(\cdot)$ | ReLU activation function |
| $A_{ij}$ | Bernoulli random variable associated with the ReLU activation function, indexed by $ij$. |
| $\rho$ | Parameter of the Bernoulli distribution associated with the ReLU unit |
| $\epsilon_r(\beta)$ | multiplier associated with paths of length $r$ in $\mathcal{N}$. |
| $\beta$ | $\frac{\rho n C}{\sqrt{\Lambda}}$. |
| $z$ | Normalization factor. |
| $\lambda_l$ | Batch normalization multiplicative factor in layer $l$. |
| $\sigma_l$ | The mean of the estimated standard deviation various elements in $\mathcal{R}(W_l^{\top}\mathcal{N}_{l-1}(x))$. |

# B  PROOFS

*Proof of Lemma 1.* There are a total of $\psi_r$ paths of length $r$ from input to output, and a total of $\Lambda^r$ unique $r$ length configurations of weights. The uniformity assumption then implies that each configuration of weights is repeated $\frac{\psi_r}{\Lambda^r}$ times. By summing over the unique configurations, and re indexing the input we arrive at Eq. 10. □

*Proof of Lemma 2.* From 12, we have that $\xi_{i_1,i_2...i_r}$ is defined as a sum of $\frac{\psi_r}{\Lambda^r}$ inputs. Since there are only $p$ distinct inputs, it holds that for each $\xi_{i_1,i_2...i_r}$ there exists a sequence $\boldsymbol{\alpha} = (\alpha_i)_{i=1}^p \in \mathbb{N}$ such that $\sum_{i=1}^d \alpha_i = \frac{\psi_r}{\Lambda^r}$, and $\xi_{i_1,i_2...i_r} = \sum_{i=1}^d \alpha_i x_i$. We, therefore, have that $\mathbb{E}[\xi_{i_1,i_2...i_r}^2] = \|\boldsymbol{\alpha}\|_2^2$. Note that the minimum value of $\mathbb{E}[\xi_{i_1,i_2...i_r}^2]$ is a solution to the following:

$$min(\mathbb{E}[\xi_{i_1,i_2...i_r}^2]) = min_{\boldsymbol{\alpha}}(\|\boldsymbol{\alpha}\|_2) \quad s.t \quad \|\boldsymbol{\alpha}\|_1 = \frac{\psi_r}{\Lambda^r}, \ (\alpha_i)_{i=1}^p \in \mathbb{N}, \tag{36}$$

which achieves its minimal value at $\forall i, \alpha_i = \frac{1}{d}\frac{\psi_r}{\Lambda^r}$. Similarly, the maximum value is achieved at $\alpha_i = \frac{\psi_r}{\Lambda^r}\delta_i$ for some index $i$. □

*Proof of Thm. 1.* We use the stirling approximation, which states $\lim_{p\to\infty}\frac{1}{p}log(\binom{p}{\alpha p})=H(\alpha)$, where $H(\alpha)=-\alpha log(\alpha)-(1-\alpha)log(1-\alpha)$. Ignoring the constants which do not depend on $\alpha$,

$$\lim_{p\to\infty}\frac{1}{p}log(\binom{p}{\alpha p}\beta^{\alpha p})=H(\alpha)+\alpha log(\beta)\tag{37}$$

which achieves its maximum value at $\alpha=\alpha^*$. □

*Proof of Thm. 2.* For brevity, we provide a sketch of the proof. It is enough to show that $\lim_{p\to\infty}\sum_{r=1}^{\alpha_1 p}\epsilon_r^2=0$ for $\beta<1$. Ignoring the constants in the binomial terms, we have:

$$\lim_{p\to\infty}\sum_{r=1}^{\alpha_1 p}\epsilon_i^2=\lim_{p\to\infty}\frac{\sum_{i=1}^{\alpha_1 p}\binom{p}{r}^2\beta^{2r}}{z^2}\le\lim_{p\to\infty}\frac{\alpha_1 p\binom{p}{\alpha_1 p}^2\beta^{2\alpha_1 p}}{z^2}\tag{38}$$

Where $z^2=\sum_{r=1}^{p}\binom{p}{r}^2\beta^{2r}$, which can be expressed using the Legendre polynomial of order $p$:

$$z^2=(1-\beta^2)^p\mathcal{P}_p(\frac{1+\beta^2}{1-\beta^2})\tag{39}$$

In order to compute the limit of Eq. 38, we use the asymptotic of the Legendre polynomial of order $p$ for $x>1$, $\mathcal{P}_p(x)\sim\frac{1}{\sqrt{2\pi p}}\frac{(x+\sqrt{x^2-1})^{p+\frac{1}{2}}}{(x^2-1)^{\frac{1}{4}}}$. For the term in the nominator of Eq. 38 , we use the Stirling approximation for factorials $p!\sim\sqrt{2\pi p}(\frac{p}{e})^p$. Substituting both approximations in Eq. 38 and taking the limit completes the proof. □

*Proof of Lemma 4.* For simplicity, we ignore the constants in the binomial coefficient, and assume $\epsilon_r=\frac{1}{z}\binom{p}{r}\beta^r$. Notice that for $\beta^*=(\frac{p}{2})$, we have that $\arg\max_r(\epsilon_r(\beta^*))=p$, $\arg\max_r(\epsilon_r(\frac{1}{\beta^*}))=1$ and $\arg\max_r(\epsilon_r(1))=\frac{p}{2}$. From the monotonicity and continuity of $\beta^r$, any value $1\ge k\ge p$ can be attained. The linear dependency $\beta(C)=\frac{\rho nC}{\sqrt{\Lambda}}$ completes the proof. □

*Proof of Thm. 3.* 1. Notice that by definition, layer $l$ is not skipped by the residual connection, and therefore $\lambda_l$ multiplies every path in the network. Therefore, $\frac{\partial\mathcal{L}_\mathcal{N}(x,\boldsymbol{w})}{\partial\lambda_l}=\frac{1}{\lambda_l}(\mathcal{L}_m(x,\boldsymbol{w})+\mathcal{L}_p(x,\boldsymbol{w}))$. Using taylor series expansion:

$$\frac{\partial\mathcal{L}_\mathcal{N}(x,\boldsymbol{w}-\mu\boldsymbol{g})}{\partial\lambda_l}\approx\frac{\partial\mathcal{L}_\mathcal{N}(x,\boldsymbol{w})}{\partial\lambda_l}-\mu\nabla_{\boldsymbol{w}}\frac{\partial\mathcal{L}_\mathcal{N}(x,\boldsymbol{w})}{\partial\lambda_l}\boldsymbol{g}\tag{40}$$

Substituting $\nabla_{\boldsymbol{w}}\frac{\partial\mathcal{L}_\mathcal{N}(x,\boldsymbol{w})}{\partial\lambda_l}=\frac{1}{\lambda_l}(\boldsymbol{g}_m+\boldsymbol{g}_p)$ in 40 we have:

$$\frac{\partial\mathcal{L}_\mathcal{N}(x,\boldsymbol{w}-\mu\boldsymbol{g}_{\boldsymbol{w}})}{\partial\lambda_l}\approx0-\mu\frac{1}{\lambda_l}(\boldsymbol{g}_m+\boldsymbol{g}_p)^\top(\boldsymbol{g}_m+\boldsymbol{g}_p)=-\mu\frac{1}{\lambda_l}\|\boldsymbol{g}_m+\boldsymbol{g}_p\|_2^2<0\tag{41}$$

And hence:

$$\lambda_l-\mu\frac{\partial\mathcal{L}_\mathcal{N}(x,\boldsymbol{w}-\mu\boldsymbol{g}_{\boldsymbol{w}})}{\partial\lambda_l}=\lambda_l+\mu^2\frac{1}{\lambda_l}\|\boldsymbol{g}_m+\boldsymbol{g}_p\|_2^2$$

$$=\lambda_l(1+\mu^2\frac{1}{\lambda_l^2}\|\boldsymbol{g}_m+\boldsymbol{g}_p\|_2^2)\tag{42}$$

Finally:

$$|\lambda_l(1+\mu^2\frac{1}{\lambda_l^2}\|\boldsymbol{g}_m+\boldsymbol{g}_p\|_2^2)|=|\lambda_l|(1+\mu^2\frac{1}{\lambda_l^2})\ge|\lambda_l|\tag{43}$$

2. Since paths of length $m$ skip layer $l$, we have that $\nabla_{\boldsymbol{w}}\frac{\partial\mathcal{L}_\mathcal{N}(x,\boldsymbol{w})}{\partial\lambda_l}=\frac{1}{\lambda_l}\boldsymbol{g}_p$. Therefore:

$$\frac{\partial\mathcal{L}_\mathcal{N}(x,\boldsymbol{w}-\mu\boldsymbol{g})}{\partial\lambda_l}\approx0-\mu\frac{1}{\lambda_l}(\boldsymbol{g}_m+\boldsymbol{g}_p)^\top\boldsymbol{g}_p=-\mu\frac{1}{\lambda_l}(\boldsymbol{g}_m^\top\boldsymbol{g}_p+\|\boldsymbol{g}_p\|_2^2)\tag{44}$$

The condition $\|\boldsymbol{g}_p\|_2>\|\boldsymbol{g}_m\|_2$ implies that $\boldsymbol{g}_m^\top\boldsymbol{g}_p+\|\boldsymbol{g}_p\|_2^2>0$, completing the proof. □

*Proof of Thm 4.* Notice that $\frac{\partial \mathcal{L}_\mathcal{N}(x, \boldsymbol{w})}{\partial C} = \frac{\partial \mathcal{L}_\mathcal{N}(x, \boldsymbol{w})}{\partial \boldsymbol{w}} \frac{\partial \boldsymbol{w}}{\partial \|\boldsymbol{w}\|_2} \sqrt{\Lambda} = \boldsymbol{g}^\top \tilde{\boldsymbol{w}} = 0$, and hence the gradient is orthogonal to the weights. We have that $\frac{\partial \mathcal{L}_\mathcal{N}(x, \boldsymbol{w})}{\partial C} = \frac{1}{C}(m\mathcal{L}_m(x, \boldsymbol{w}) + p\mathcal{L}_p(x, \boldsymbol{w}))$. Using taylor series expansion we have:

$$\frac{\partial \mathcal{L}_\mathcal{N}(x, \boldsymbol{w} - \mu \boldsymbol{g})}{\partial C} \approx \frac{\partial \mathcal{L}_\mathcal{N}(x, \boldsymbol{w})}{\partial C} - \mu \nabla_{\boldsymbol{w}} \frac{\partial \mathcal{L}_\mathcal{N}(x, \boldsymbol{w})}{\partial C} \boldsymbol{g} \quad (45)$$

For the last term we have:

$$\nabla_{\boldsymbol{w}} \frac{\partial \mathcal{L}_\mathcal{N}(x, \boldsymbol{w})}{\partial C} \boldsymbol{g} = (m\mathcal{L}_m(x, \boldsymbol{w}) + p\mathcal{L}_p(x, \boldsymbol{w})) \nabla_{\boldsymbol{w}} \frac{\sqrt{\Lambda}}{\|\boldsymbol{w}\|_2} \boldsymbol{g} + \frac{1}{C}(m\boldsymbol{g}_m + p\boldsymbol{g}_p)^\top \boldsymbol{g}$$

$$= (m\mathcal{L}_m(x, \boldsymbol{w}) + p\mathcal{L}_p(x, \boldsymbol{w})) \frac{\boldsymbol{w}^\top \boldsymbol{g}}{C^{\frac{3}{2}}} + \frac{1}{C}(m\boldsymbol{g}_m + p\boldsymbol{g}_p)^\top \boldsymbol{g} = \frac{1}{C}(m\boldsymbol{g}_m + p\boldsymbol{g}_p)^\top \boldsymbol{g}, \quad (46)$$

where the last step stems from the fact that $\boldsymbol{w}^\top \boldsymbol{g} = 0$. Substituting $\nabla_{\boldsymbol{w}} \frac{\partial \mathcal{L}_\mathcal{N}(x, \boldsymbol{w})}{\partial C} = \frac{1}{C}(m\boldsymbol{g}_m + p\boldsymbol{g}_p)$ in 45 we have:

$$\frac{\partial \mathcal{L}_\mathcal{N}(x, \boldsymbol{w} - \boldsymbol{\mu g_w})}{\partial C} \approx 0 - \mu \frac{1}{C}(m\boldsymbol{g}_m + p\boldsymbol{g}_p)^\top (\boldsymbol{g}_m + \boldsymbol{g}_p)$$

$$= -\mu \frac{1}{C}(m\|\boldsymbol{g}_p\|_2^2 + p\|\boldsymbol{g}_p\|_2^2 + (m+p)\boldsymbol{g}_p^\top \boldsymbol{g}_m) \quad (47)$$

$\square$

*Proof of Thm 5.* Inserting Eq. 31 into Eq. 33 we have that:

$$\theta_k(\mathbb{R}, \boldsymbol{\epsilon}) = \frac{1}{2} log(\frac{\sum_{r=2}^p \epsilon_r^2 r(r-1)}{\sum_{r=2}^p \epsilon_r^2 r}) - \frac{\sum_{r=2}^p \epsilon_r^2 r(r-2)}{\sum_{r=2}^p \epsilon_r^2 r^2} \quad (48)$$

We denote the matrices $V'$ and $V''$ such that $V'_{ij} = r\delta_{ij}$ and $V''_{ij} = r(r-1)\delta_{ij}$. We then have:

$$\theta_k(\mathbb{R}, \boldsymbol{\epsilon}) = \frac{1}{2} log(\frac{\boldsymbol{\epsilon}^\top V'' \boldsymbol{\epsilon}}{\boldsymbol{\epsilon}^\top V' \boldsymbol{\epsilon}}) - \frac{\boldsymbol{\epsilon}^\top (V'' - V')\boldsymbol{\epsilon}}{\boldsymbol{\epsilon}^\top (V'' + V')\boldsymbol{\epsilon}} \quad (49)$$

$$max_{\boldsymbol{\epsilon}} \theta_k(\mathbb{R}, \boldsymbol{\epsilon}) \leq max_{\boldsymbol{\epsilon}}(\frac{1}{2} log(\frac{\boldsymbol{\epsilon}^\top V'' \boldsymbol{\epsilon}}{\boldsymbol{\epsilon}^\top V' \boldsymbol{\epsilon}})) - min_{\boldsymbol{\epsilon}}(\frac{\boldsymbol{\epsilon}^\top (V'' - V')\boldsymbol{\epsilon}}{\boldsymbol{\epsilon}^\top (V'' + V')\boldsymbol{\epsilon}})$$

$$= \frac{1}{2} log\left(max_i(V''_{ii} V'^{-1}_{ii})\right) - min_i\left((V''_{ii} - V'_{ii})(V''_{ii} + V'_{ii})^{-1}\right)$$

$$= \frac{1}{2} log(p-1) - (1 - \frac{2}{p}) = \theta_k(\mathbb{R}, \boldsymbol{\epsilon}^*) \quad (50)$$

$\square$

## C  ADDITIONAL EXPERIMENTS

Fig. 1(d) and 1(e) report the experimental results of a straightforward setting, in which the task is to classify a mixture of 10 multivariate Gaussians in 50D. The input is therefore of size 50. The loss employed is the cross entropy loss of ten classes. The network has 10 blocks, each containing 20 hidden neurons, a batch normalization layer, and a skip connection. Training was performed on 10,000 samples, using SGD with minibatches of 50 samples.

Next, we provide additional experiments performed on the public CIFAR-10 and CIFAR-100 data sets (Krizhevsky, 2009). The public ResNet code of `https://github.com/facebook/fb.resnet.torch` is used for networks of depth 32.

As noted in Sec. 4.2, the dynamic behavior can be present in the Batch Normalization multiplicative coefficient or in the weight matrices themselves. In the following experiments, it seems that

until the learning rate is reduced, the dynamic behavior is manifested in the Batch Normalization multiplicative coefficients and then it moves to the convolution layers themselves. We therefore absorb the BN coefficients into the convolutional layer using the public code of `https://github.com/e-lab/torch-toolbox/tree/master/BN-absorber`. Note that the multiplicative coefficient of Batch Normalization is typically refereed to as $\gamma$. However, throughout our paper, since we follow the notation of Choromanska et al. (2015a), $\gamma$ refers to the number of paths. The multiplicative factor of Batch normalization appears as $\lambda$ in Sec. 4.

Fig. 2 depicts the results. There are two types of plots: Fig. 2(a,c) presents for CIFAR-10 and CIFAR-100 respectively the magnitude of the various convolutional layers for multiple epochs (similar in type to Fig. 1(d) in the paper). Fig. 2(b,d) depict for the two datasets the mean of these norms over all convolutional layers as a function of epoch (similar to Fig. 1(e)).

As can be seen, the dynamic phenomenon we describe is very prominent in the public ResNet implementation when applied to these conventional datasets: the dominance of paths with fewer skip connections increases over time. Moreover, once the learning rate is reduced in epoch 81 the phenomenon we describe speeds up.

In Fig. 3 we present the multiplicative coefficient of the Batch Normalization when not absorbed. As future work, we would like to better understand why these coefficients start to decrease once the learning rate is reduced. As shown above, taking the magnitude of the convolutions into account, the dynamic phenomenon we study becomes even more prominent at this point. The change of location from the multiplicative coefficient of the Batch Normalization layers to the convolutions themselves might indicate that Batch Normalization is no longer required at this point. Indeed, Batch Normalization enables larger training rates and this shift happens exactly when the training rate is reduced. A complete analysis is left for future work.

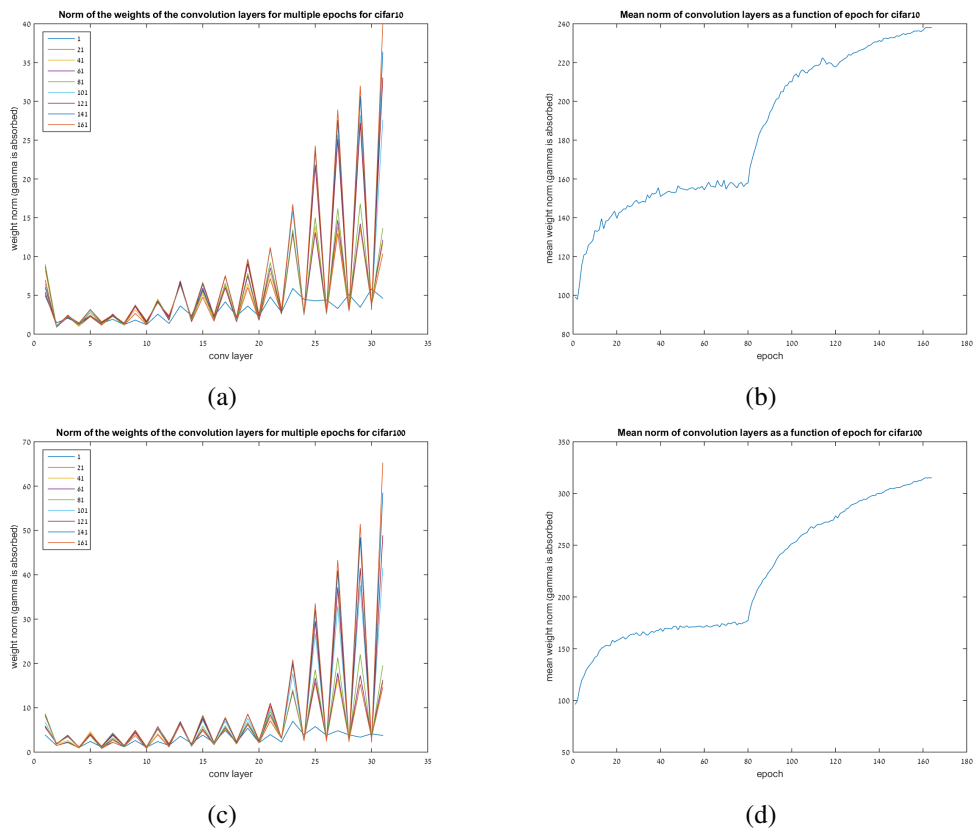

Figure 2: (a,c) The Norm of the convolutional layers once the factors of the subsequent Batch Normalization layers are absorbed, shown for CIFAR-10 and CIFAR-100 respectively. Each graph is a different epoch, see legend. Waving is due to the interleaving architecture of the convolutional layers. (b,d) Respectively for CIFAR-10 and CIFAR-100, the mean of the norm of the convolutional layers' weights per epoch.

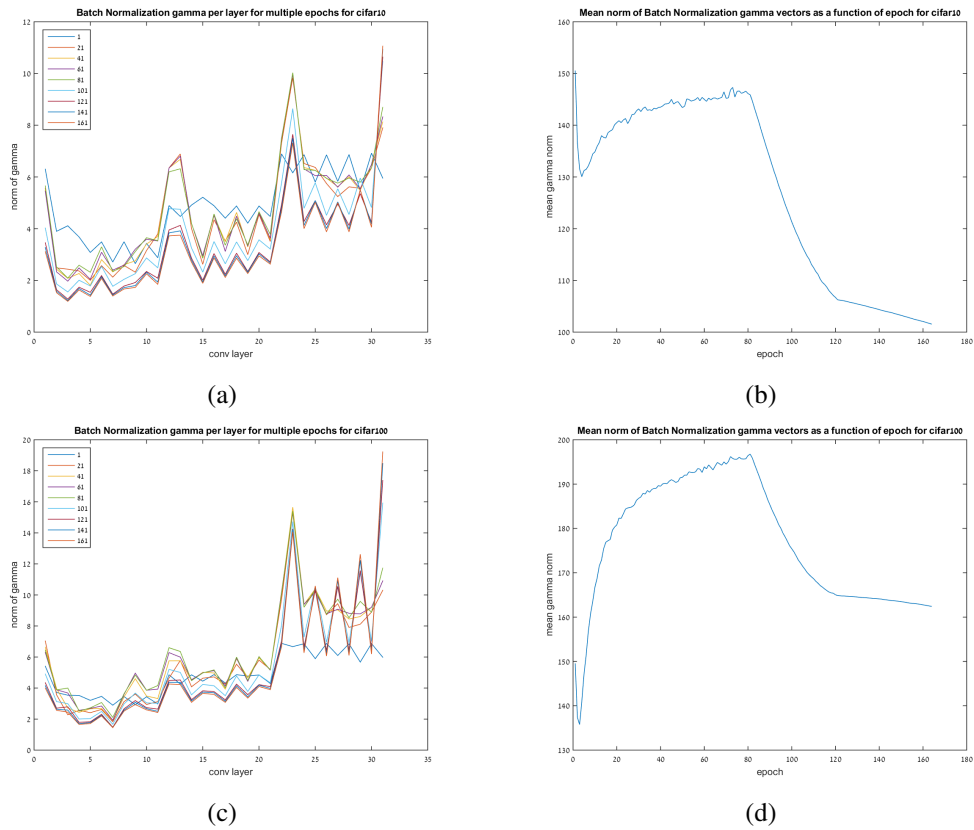

(a)

(b)

(c)

(d)

Figure 3: The norms of the multiplicative Batch Normalization coefficient vectors. (a,c) The Norm of the coefficients, shown for CIFAR-10 and CIFAR-100 respectively. Each graph is a different epoch (see legend). Since there is no monotonic increase between the epochs in this graph, it is harder to interpret. (b,d) Respectively for CIFAR-10 and CIFAR-100, the mean of the norm of the multiplicative factors per epoch.

