# Peer review of "The loss surface of residual networks: Ensembles and the role of batch normalization"

_ICLR 2017 — rejected_

[Reviewer Comment · AnonReviewer3 · 04 Dec 2016]
**assumptions and claims**

-  The issue regarding unrealistic assumptions of spin glass analysis for landscape of neural networks was posed as an open problem in COLT 2015.[1] Can you discuss the effect of this problem for your analysis?

- Regarding assumption that minimal of (12) should hold, is this assumption is realistic or not?

-Choromanska(AISTATS 2015) supports the claims based on theoretical results with many empirical results however you do not provide such analysis. Can you support your claims with reasonable empirical results?

- What is the empirical setup for Figure 1?

-Fractal net paper on arxiv claim that residuals are incidental. Can you elaborate on that based on your framework? What about densely connected conv networks(huang,2016 )? 

[1] A. Choromanska, Y. LeCun, G. Ben Arous, Open Problem: The landscape of the loss surfaces of multilayer networks, in the Conference on Learning Theory (COLT), Open Problems, 2015

[Author Response · Lior Wolf · 16 Dec 2016]
**More runs**

We looked back at the comment of AnonReviewer3. We are the first to analyze the dynamic behavior of ResNets during training and point to a novel phenomenon, which we analyze theoretically. We reveal both the underlying reason for this phenomenon and its profound effect on the training process. However, we might not have demonstrated convincingly enough the existence of this phenomenon on conventional datasets. We therefore ran a few last minute experiments to show exactly this.

We took the ResNet code of

[Official Review · AnonReviewer2 · rating 7 · confidence 3 · 17 Dec 2016]
**Interesting theoretical analysis (with new supporting experiments) but presented in a slightly confusing fashion.**

Summary:
In this paper, the authors study ResNets through a theoretical formulation of a spin glass model. The conclusions are that ResNets behave as an ensemble of shallow networks at the start of training (by examining the magnitude of the weights for paths of a specific length) but this changes through training, through which the scaling parameter C (from assumption A4) increases, causing it to behave as an ensemble of deeper and deeper networks.

Clarity:
This paper was somewhat difficult to follow, being heavy in notation, with perhaps some notation overloading. A summary of some of the proofs in the main text might have been helpful.

Specific Comments:
- In the proof of Lemma 2, I'm not sure where the sequence beta comes from (I don't see how it follows from 11?)

- The ResNet structure used in the paper is somewhat different from normal with multiple layers being skipped? (Can the same analysis be used if only one layer is skipped? It seems like the skipping mostly affects the number of paths there are of a certain length?)

- The new experiments supporting the scale increase in practice are interesting! I'm not sure about Theorems 3, 4 necessarily proving this link theoretically however, particularly given the simplifying assumption at the start of Section 4.2?

[Author Response · Etai Littwin · 18 Dec 2016]
**Title: revised version**

We just revised our manuscript based on the very meaningful discussions we had with the reviewing team. For convenience, the changes are marked in red.

In addition to incorporating everything we have promised so far and adding discussion based on the reviewers’ suggestions, we were able to relax the assumptions used for the results of Sec.  4. This change addresses a concern raised by AnonReviewer3 and was done by using the assumption-free expression for the output of the network in Eq. 9 in order to compute the loss in Lemma 3.

Overall, believe that what can be considered as the main results of our manuscript, i.e., the dynamic behavior during the training of ResNets, is now essentially assumption free and is also well supported experimentally. We thank the reviewers for their crucial role in improving the manuscript.

[Official Review · AnonReviewer3 · rating 7 · confidence 3 · 20 Dec 2016 (modified: 23 Jan 2017)]
**promising insightful results**

This paper extend the Spin Glass analysis of Choromanska et al. (2015a) to Res Nets which yield the novel dynamic ensemble results for Res Nets and the connection to Batch Normalization and the analysis of their loss surface of Res Nets.

The paper is well-written with many insightful explanation of results. Although the technical contributions extend the Spin Glass model analysis of the ones by Choromanska et al. (2015a), the updated version could eliminate one of the unrealistic assumptions and the analysis further provides novel dynamic ensemble results and the connection to Batch Normalization that gives more insightful results about the structure of Res Nets. 

It is essential to show this dynamic behaviour in a regime without batch normalization to untangle the normalization effect on ensemble feature. Hence authors claim that steady increase in the L_2 norm of the weights will maintain the this feature but setting for Figure 1 is restrictive to empirically support the claim. At least results on CIFAR 10 without batch normalization for showing effect of L_2 norm increase and results that support claims about Theorem 4 would strengthen the paper.

This work provides an initial rigorous framework to analyze better the inherent structure of the current state of art Res Net architectures and its variants which can stimulate potentially more significant results towards careful understanding of current state of art models (Rather than always to attempting to improve the performance of Res Nets by applying intuitive incremental heuristics, it is important to progress on some solid understanding too).

[Official Review · AnonReviewer1 · rating 3 · confidence 5 · 20 Dec 2016]
**interesting extension of the result of Choromanska et al. but too incremental**

This paper shows how spin glass techniques that were introduced in Choromanska et al. to analyze surface loss of deep neural networks can be applied to deep residual networks. This is an interesting contribution but it seems to me that the results are too similar to the ones in Choromanska et al. and thus the novelty is seriously limited. Main theoretical techniques described in the paper were already introduced and main theoretical results mentioned there were in fact already proved. The authors also did not get rid of lots of assumptions from Choromanska et al. (path-independence, assumptions about weights distributions, etc.).

[Reviewer Comment · AnonReviewer1 · 20 Jan 2017]
**final evaluation**

Authors' responses did not make me change my mind. I still think it is a clear rejection paper.

[Final Decision · Program Chairs · 06 Feb 2017]
**ICLR committee final decision**

The paper presents an analysis of residual networks and argues that the residual networks behave as ensembles of shallow networks, whose depths are dynamic. The authors argue that their model provides a concrete explanation to the effectiveness of resnets. 
 
 However, I have to agree with reviewer 1 that the assumption of path independence is deeply flawed. In my opinion, it was also flawed in the original paper. Using that as a justification to continue this line of research is not the right approach. We cannot construct a single practical scenario where path independence may be expected to hold. So we should not be encouraging papers to continue this line of flawed reasoning.
 
 I thus cannot recommend acceptance of this paper.